

# Impact of dialysis modality conversion on the health-related quality of life of peritoneal dialysis patients: a retrospective cohort study in China

Heqi Sun[1,3,*], Ye Zhuang[1,3,*], Lanying Gao[2], Ningze Xu[1], Yan Xiong[2], Min Yuan[1,3], Jun Lu[1,3] and Jianming Ye[2]

[1] School of Public Health, Fudan University, Shanghai, China
[2] Department of Nephrology, The First People's Hospital of Kunshan, Suzhou, China
[3] China Research Center on Disability, Fudan University, Shanghai, China
[*] These authors contributed equally to this work.

Corresponding authors
Jun Lu, lujun@shmu.edu.cn
Jianming Ye, ks_yjm@163.com

## ABSTRACT

**Background**. To analyze the health-related quality of life associated with the conversion of dialysis modality among end-stage renal disease patients in China.

**Methods**. Patients were recruited from hospitals and a dialysis center in Kunshan, China. Patients converting from continuous ambulatory peritoneal dialysis to automated peritoneal dialysis were recruited as the observation group ($n = 64$), and patients continuing with continuous ambulatory peritoneal dialysis treatment were included in the control group ($n = 64$) after matching in this retrospective cohort study. Their health-related quality of life was measured using the kidney disease quality of life instrument in 2019 and 2020, respectively. Baseline socio-demographic characteristics and clinical data were collected in 2019. The before-and-after cross-group comparisons of subscale scores of two groups were conducted using a Student's t-test. Multiple linear regression models were fitted to identify the factors associated with the change of each scale.

**Results**. The health-related quality of life scores of the two groups was comparable in baseline, while the observation group had higher scores in Physical Component Summary ($51.92 \pm 7.50$), Kidney Disease Component Summary ($81.21 \pm 8.41$), Symptoms ($90.76 \pm 6.30$), Effects ($82.86 \pm 11.42$), and Burden ($69.04 \pm 15.69$) subscales after one year. In multivariate regression analysis, the change of Physical Component Summary was significantly associated with conversion to APD ($\beta = 11.54$, 95% CI [7.26–15.82]); the change of Mental Component Summary with higher education ($\beta = -5.96$, 95% CI [$-10.18$–$-1.74$]) and CCI ($>2$) ($\beta = 5.39$, 95% CI [1.05–9.73]); the change of Kidney Disease Component Summary with conversion to APD ($\beta = 15.95$, 95% CI [10.19–21.7]) and age ($>60$ years) ($\beta = -7.36$, 95% CI [$-14.11$–$-0.61$]); the change of Symptoms with CCI ($>2$) ($\beta = 7.96$, 95% CI [1.49–14.44]); the change of Effects with conversion to APD ($\beta = 19.23$, 95% CI [11.57–26.88]); and the change of Burden with conversion to APD ($\beta = 22.40$, 95% CI [13.46–31.34]), age ($>60$ years) ($\beta = -12.12$, 95% CI [$-22.59$–$-1.65$]), and higher education ($\beta = -10.38$, 95% CI [$-19.79$–$-0.98$]).

**Conclusions**. The conversion of dialysis modality had a significant impact on the scores of most subscales. Patients converting from continuous ambulatory peritoneal dialysis

to automated peritoneal dialysis generally had improved health-related quality of life scores.

## INTRODUCTION

Renal transplantation, hemodialysis (HD) and peritoneal dialysis (PD) are three common renal replacement therapy (RRT) options for patients with end-stage renal disease (ESRD). As the clinical concept of "PD First" has been widely recognized globally, many countries have increasingly practiced PD instead of HD in clinical settings mainly because it has less impact on patients' daily life routines (*Rabindranath et al., 2007*) and saves their healthcare costs (*Vaccaro & Sopranzi, 2017*; *Julián et al., 2013*).

Among major modalities of PD, ambulatory peritoneal dialysis (CAPD) and automated peritoneal dialysis (APD) are the most one. CAPD is performed manually, whereas APD is performed with the assistance of mechanical device and has been associated with better patient compliance (*Bernardini, Nagy & Piraino, 2000*). Since many systematic reviews found that the safety and clinical outcomes of CAPD and APD are largely comparable (*Rabindranath et al., 2007*; *Athanasios et al., 2020*; *Tang, Chen & Fang, 2016*), evidence on health-related quality of life (HRQOL) is important when nephrologists and patients choose PD modality in practice (*Wright & Wilson, 2015*). To the best of our knowledge, studies conducted in other countries showed that the HRQOL was largely equivalent between patients treated with CAPD and APD (*Guney et al., 2010*; *Michels et al., 2011*; *Yang et al., 2018*; *Mpharm et al., 2020*). Since these studies were mainly cross-sectional designs that could not analyze the impact of converting PD modalities on ESRD patients, current knowledge of this topic remains little. Besides, some Chinese studies have targeted the HRQOL of HD and PD patients (*Zhang et al., 2007*; *Bieber et al., 2014*), but they did not conclude on the HRQOL of CAPD and APD patients. Therefore, this study fills the gaps of knowledge and would provide real-world evidence for the selection of dialysis modalities in clinical settings.

In mainland China, it was estimated that about 1.5 million patients need dialysis. However, PD accounts for less than 15%, of which CAPD is the main modality (*Chinese National Renal Data System, 2021*). Research regarding APD remains to be almost blank. Fortunately, health insurance authorities and healthcare facilities in Kunshan, Jiangsu, China have tried to promote the use of APD since 2019. Kunshan is a highly developed city in China with a GDP per capita that is equivalent to a city in a developed country. To control the impact of socio-economic status on HRQOL, Kunshan was selected as the sample city while ensuring the comparability of our findings with international literature. Therefore, this study is the first publication comparing the one-year change in HRQOL of CAPD and APD patients in China and identifying its influencing factors.

## METHODS

### Design and ethical statement

This observational cohort study was designed to investigate the association between the conversion of dialysis modality and health-related quality of life among the Chinese end-stage renal disease patients. All procedures performed in this study involving human participants were in accordance with the Declaration of Helsinki (as revised in 2013). The study was approved by the Medical Research Ethics Committee of School of Public Health, Fudan University (IRB approval number 2020-08-0844). Written informed consent was obtained from all participants at the recruitment stage of the study.

### Study setting and subjects

Patients were recruited from the hospitals and a dialysis center in Kunshan approached by trained interviewers. The inclusion criteria were: (1) a clinical diagnosis of ESRD; (2) aged 18 years or older; (3) received CAPD/APD for more than 3 months; (4) able to communicate verbally; (5) provide informed consent.

Kunshan has introduced home-based APD services since 2019, with supporting medical insurance reimbursement policies. Through interviews with patients and medical staff, the research team learned that most patients who chose to convert from CAPD to APD were mainly in the consideration of social and economic reasons: Socially, APD helps patients with daily activities, work, and study throughout the day; Economically, the individual payment proportion of choosing APD is lower, and the economic burden of dialysis was alleviated.

This study included all 67 eligible ESRD patients converting from CAPD to APD as the observation group (hereinafter referred to as APD group), of which 64 patients completely responded to the interviews, resulting in a response rate of 95.52%. Meanwhile, we also interviewed 135 ESRD patients continuing with CAPD treatment, of which 125 patients completely responded, and the response rate was 92.59%.

After matching was conducted based on dialysis vintage, age, and gender, 64 ESRD patients continuing with CAPD treatment were included in the control group (hereinafter referred to as CAPD group). During the matching process, completely matched patients were included first. Incompletely matched patients were included by the order of dialysis vintage, age, and gender. If more than one patient met the matching criteria, the first patient on the matching list was included. We estimated that this sample size of 128 (64 in each group) would provide 99% power at a two-tailed 5% significance level by using G*Power analysis.

### Procedures and assessment methods

In this retrospective study, we collected baseline data of the observation group and control group in August 2019 and then remeasured the HRQOL of two groups in August 2020.

### HRQOL instruments

HRQOL data were collected using the Kidney Disease and Quality of Life-36 (KDQOL-36) (*Rand, 2021*). KDQOL-36 contains a Short Form-12 (SF-12) and three kidney disease-specific subscales, in which there are four items for the burden of kidney disease, 12 items
for symptoms/problems, and eight items for effects of kidney disease. The kidney disease component summary (KDCS) is scored by averaging the three disease-specific subscales (*i.e.,* symptoms, effects, and burden). The Physical Component Summary (PCS) and Mental Component Summary (MCS) can be calculated from SF-12. All scores range from 0 to 100, with higher scores representing better HRQOL.

KDQOL-36 has been validated and proved reliable for the measurement of HRQOL in Chinese ESRD patients (*Tao, Chow & Wong, 2014*). *Pan et al. (2018)* also employed KDQOL-36 to measure the HRQOL of ESRD patients in Eastern China, proving the effectiveness of KDQOL-36 in the sample population.

To measure and compare the one-year differences between two groups in 6 subscales (*i.e.,* PCS, MCS, KDCS, symptoms, effects, and burden), we defined these differences as △PCS, △MCS, △KDCS, △symptoms, △effects, and △burden, respectively.

### Sociodemographic and clinical data

In this study, sociodemographic and clinical data were also collected. Socio-demographic characteristics were self-reported, including age, gender, marital status, education level, employment status, and dependency status. Clinical factors were retrieved from medical records, containing dialysis characteristics (dialysis vintage and Kt/V), Charlson Comorbidity Index (CCI), body mass index (BMI), and biochemical data (blood albumin and hemoglobin).

### Statistical analysis

The socio-demographic and clinical characteristics were compared between CAPD and APD groups using chi-square tests. The before-and-after cross-group comparisons of the changes in scores of six HRQOL subscales were conducted using a *T*-test.

In univariate linear regression and the subsequent multivariate linear regression, we assessed the associations between sociodemographic and clinical characteristics and the changes in HRQOL scores. The continuous dependent variables were the changes in 6 subscales (△PCS, △MCS, △KDCS, △symptoms, △effects, and △burden). All continuous independent variables were coded into categorical variables based on their clinically meaningful values (*Jia et al., 2015*) or means/medians concerning that the associations may not be linear. The categorical variables were PD modality (continue with CAPD/convert to APD), age ($\leq$60 years />60 years), gender (male/female), marital status (with spouse/without spouse), education level (lower (no formal/primary/secondary school)/higher(high school/above)), employment status (employed/unemployed (including retired)), dependency status (self-care/assisted), dialysis vintage (<4 years/ $\geq$4 years), Kt/V($\leq$1.7/>1.7), CCI($\leq$2/>2), BMI level (low ($\leq$25 kg/m2)/ high (>25 kg/m2)), albumin level (low (<37 g/l)/high ($\geq$37 g/l)), hemoglobin level (low (<11 g/dl)/high ($\geq$11 g/dl)).

Dialysis-related variables were included with references to previous literature (*Yang et al., 2018*; *Pan et al., 2018*). All categorical variables were entered into six different multiple linear regression analyses to explore their associations with the changes in HRQOL subscale scores independently, regardless of their statistical significances in the univariate analyses,
to assess the impact of dialysis modality conversion on the changes in HRQOL scores after adjusting for socio-demographic characteristics and clinical indicators. All analyses were performed using SPSS 25.0, with a *p-value* of less than 0.05 being considered as statistically significant.

## RESULTS

A total of 128 PD patients were included in this study, with 64 of the patients continuing with CAPD and 64 of whom converting to APD. Their characteristics are displayed in Table 1. The mean (SD) age was 52.95(13.48) years, with 53.13% male and the mean (SD) dialysis vintage was 4.10(3.34) years. There were significant differences in dependency status between the two groups. Other characteristics related to PD technique, such as residual diuresis and daily ultrafiltration, were not different between the two studied groups.

Mean (SD) PCS, MCS, KDCS, symptoms, effects, and burden scores for the CAPD and APD groups in two years are shown in Table 2 respectively. There were no significant differences between the CAPD group and APD group in the baseline. However, by the year 2020, PCS, KDCS, Symptoms, Effects, and Burden scores of the APD group were significantly higher than the CAPD group.

Table 3 shows the results of the univariate analysis. In univariate analysis, we found that many factors associated with the changes in HRQOL scores were statistically significant.

In multivariate analysis (Table 4), after adjusting for the effects of socio-demographic and clinical factors in six models, the following factors were significantly associated with increasing the of HRQOL scores: conversion to APD with △PCS; higher education and CCI (>2) with △MCS; conversion to APD and age (>60 years) with △KDCS; CCI (>2) with △Symptoms; conversion to APD with △Effects; and conversion to APD, age (>60 years), and higher education with △Burden.

## DISCUSSION

In this study, ESRD patients had higher scores in PCS, KDCS, Symptoms, Effects, and Burden subscales after converting to APD. In multivariate regression analysis, factors significantly associated with the improvement of HRQOL scores were: conversion to APD, ≤60 years of age, lower education, and higher hemoglobin. These results suggest evidence for the effectiveness of APD among Chinese ESRD patients and guide the strategy of selecting appropriate dialysis techniques.

HRQOL was compared between the APD group and CAPD group, and no statistical difference was found at baseline. However, after one year of treatment, the physiological and disease-specific subscale scores of APD patients were significantly higher than those of CAPD patients in 2020. Similar results were also reported in Singapore (*Yang et al., 2018*) and Mexico (*Cortés-Sanabria et al., 2013*). Yet Wit et al. found no significant differences in physical health between APD and CAPD patients (*Wit et al., 2001*). This difference in results may be due to the fact that Wit et al. used SF-36 and EQ-5D instead of KDQOL-36. In addition to the fact that APD operated at night can liberate patients during the day,

**Table 1 Baseline characteristics of sample participants.**

| Characteristic | Total ($n = 128$) | CAPD ($n = 64$) | APD ($n = 64$) | p value |
|---|---|---|---|---|
| **Age(year), mean (SD)** | 52.95(13.48) | 52.88(13.38) | 53.02(13.69) | 0.95 |
| **Gender, n (%)** | | | | 0.48 |
| Male | 68(53.13) | 32(50.00) | 36(56.25) | |
| Female | 60(48.87) | 32(50.00) | 28(43.75) | |
| **Marital status, n (%)** | | | | 1.00 |
| With spouse | 112(87.50) | 56(87.50) | 56(87.50) | |
| Without spouse | 16(12.50) | 8(12.50) | 8(12.50) | |
| **Education level, n (%)** | | | | 0.14 |
| Lower (illiterate/primary/secondary) | 82(64.60) | 37(57.81) | 45(70.31) | |
| Higher (high school/above) | 46(87.50) | 27(42.19) | 19(29.69) | |
| **Employment status, n (%)** | | | | 0.34 |
| Employed | 41(32.03) | 18(28.12) | 23(35.94) | |
| Unemployed & retired | 87(67.97) | 46(71.88) | 41(64.06) | |
| **Dependency status, n (%)** | | | | **<0.001** |
| Self-care | 83(64.84) | 54(84.38) | 29(45.31) | |
| Assisted | 45(35.16) | 10(15.62) | 35(54.69) | |
| **Dialysis** | | | | |
| Dialysis vintage (year), mean (SD) | 4.10(3.34) | 4.09(3.33) | 4.11(3.38) | 0.98 |
| Kt/V, mean (SD) | 1.80(0.57) | 1.76(0.55) | 1.84(0.59) | 0.46 |
| **Clinical** | | | | |
| CCI, mean (SD) | 2.55(1.00) | 2.55(1.01) | 2.55(0.99) | 1.00 |
| BMI (kg/m$^2$), mean (SD) | 22.82(3.29) | 22.67(3.33) | 22.97(3.27) | 0.60 |
| Albumin (g/L), mean (SD) | 36.99(4.13) | 37.6(3.16) | 36.39(4.86) | 0.10 |
| Hemoglobin (g/L), mean (SD) | 10.69(1.81) | 10.91(1.88) | 10.47(1.73) | 0.17 |

**Notes.**
Values in bold indicate $P < 0.05$.
CAPD, continuous ambulatory peritoneal dialysis; APD, automated peritoneal dialysis; CCI, Charlson Comorbidity Index; BMI, body mass index.
Residual diuresis and daily ultrafiltration were not reported in the table because they were not significantly different between two groups.

**Table 2 HRQOL scores in different group.**

| | 2019 | | | | 2020 | | | |
|---|---|---|---|---|---|---|---|---|
| | CAPD ($n = 64$) | APD ($n = 64$) | t | p value | CAPD ($n = 64$) | APD ($n = 64$) | t | p value |
| **PCS** | 39.02 ± 8.77 | 40.37 ± 8.26 | −0.90 | 0.37 | 39.21 ± 8.70 | 51.92 ± 7.50 | −8.85 | **<0.001** |
| **MCS** | 45.50 ± 9.38 | 46.57 ± 8.23 | −0.69 | 0.49 | 51.80 ± 7.59 | 53.97 ± 7.33 | −1.65 | 0.10 |
| **KDCS** | 51.35 ± 15.54 | 50.43 ± 13.05 | 0.36 | 0.72 | 62.83 ± 9.85 | 81.21 ± 8.41 | −11.34 | **<0.001** |
| **Symptoms** | 76.20 ± 16.47 | 73.11 ± 12.86 | 1.18 | 0.24 | 86.17 ± 11.36 | 90.76 ± 6.30 | −2.83 | **0.01** |
| **Effects** | 56.74 ± 19.02 | 55.32 ± 14.89 | 0.47 | 0.64 | 61.91 ± 10.55 | 82.86 ± 11.42 | −10.78 | **<0.001** |
| **Burden** | 21.09 ± 19.81 | 22.85 ± 18.94 | −0.51 | 0.61 | 40.43 ± 16.55 | 69.04 ± 15.69 | −10.04 | **<0.001** |

**Notes.**
Values in bold indicate $P < 0.05$.
CAPD, continuous ambulatory peritoneal dialysis; APD, automated peritoneal dialysis; PCS, physical component summary; MCS, mental component summary; KDCS, kidney disease component summary.

**Table 3** Univariate linear regression analysis of the change of HRQOL scores.

| Variable (reference group) | ΔPCS | ΔMCS | ΔKDCS | ΔSymptoms | ΔEffects | ΔBurden |
|---|---|---|---|---|---|---|
| **PD modality (continue with CAPD)** | | | | | | |
| convert to APD | **11.35** | 1.20 | **19.29** | **7.68** | **22.37** | **26.86** |
| *p* value | **<0.001** | 0.57 | **<0.001** | **0.01** | **<0.001** | **<0.001** |
| **Age (≤60 years)** | | | | | | |
| >60 | 0.98 | **4.64** | −2.02 | 3.25 | −0.30 | −8.38 |
| *p* value | 0.70 | **0.04** | 0.58 | 0.34 | 0.95 | 0.13 |
| **Gender (male)** | | | | | | |
| Female | −1.65 | −0.83 | −2.65 | 1.73 | −4.12 | −6.56 |
| *p* value | 0.45 | 0.67 | 0.40 | 0.55 | 0.30 | 0.16 |
| **Marital status (with spouse)** | | | | | | |
| Without spouse | 0.93 | −4.55 | −3.45 | −7.44 | −2.96 | −0.50 |
| *p* value | 0.78 | 0.11 | 0.47 | 0.09 | 0.62 | 0.94 |
| **Education level (lower)** | | | | | | |
| Higher | −4.97 | **−5.84** | **−7.08** | **−6.70** | −4.43 | **-9.36** |
| *p* value | 0.03 | **<0.001** | **<0.001** | **0.03** | 0.28 | **0.05** |
| **Employment status (employed)** | | | | | | |
| Unemployed | −0.10 | −3.66 | 1.13 | −5.06 | 0.38 | 8.76 |
| *p* value | 0.97 | 0.07 | 0.74 | 0.10 | 0.93 | 0.08 |
| **Dependency status (self-care)** | | | | | | |
| Assisted | 3.75 | 1.48 | **12.63** | **11.34** | **13.75** | **13.52** |
| *p* value | 0.10 | 0.46 | **<0.001** | **<0.001** | **<0.001** | **0.01** |
| **Dialysis vintage (<4 years)** | | | | | | |
| ≥4 | 3.23 | 0.74 | 3.99 | 2.26 | 4.31 | 4.41 |
| *p* value | 0.13 | 0.70 | 0.20 | 0.44 | 0.28 | 0.35 |
| **Kt/V (≤1.7)** | | | | | | |
| >1.7 | −2.26 | 1.04 | 1.91 | −0.22 | 5.47 | 1.54 |
| *p* value | 0.30 | 0.59 | 0.55 | 0.94 | 0.17 | 0.74 |
| **CCI (≤2)** | | | | | | |
| >2 | −0.41 | **4.54** | 6.39 | **10.61** | 6.44 | 2.84 |
| *p* value | 0.86 | **0.02** | 0.06 | **0.001** | 0.13 | 0.57 |
| **BMI (≤25 kg/m²)** | | | | | | |
| >25 | 2.76 | −1.42 | −0.50 | 3.29 | −1.65 | −2.51 |
| *p* value | 0.29 | 0.54 | 0.90 | 0.35 | 0.73 | 0.66 |
| **Albumin (<37 g/L)** | | | | | | |
| ≥37 | −1.78 | −3.04 | **−8.72** | **−8.20** | **−11.99** | −6.83 |
| *p* value | 0.41 | 0.11 | **0.01** | **0.01** | **0.01** | 0.15 |
| **Hemoglobin (<11 g/L)** | | | | | | |
| ≥11 | −2.94 | **3.61** | −1.31 | −1.82 | −2.58 | −0.66 |
| *p* value | 0.18 | **0.01** | 0.70 | 0.53 | 0.52 | 0.89 |

Notes.

Values in bold indicate *P* < 0.05.

CAPD, continuous ambulatory peritoneal dialysis; APD, automated peritoneal dialysis; CCI, Charlson Comorbidity Index; BMI, body mass index; PCS, physical component summary; MCS, mental component summary; KDCS, kidney disease component summary.

Residual diuresis and daily ultrafiltration were not reported in the table because they were not correlated in univariate analysis with the HRQOL scores.

**Table 4  Multiple linear regression analysis of the change of HRQOL scores.**

| Variable | Reference | ΔPCS β (95% CI) | p | ΔMCS β (95% CI) | p | ΔKDCS β (95% CI) | p | ΔSymptoms β (95% CI) | p | ΔEffects β (95% CI) | p | ΔBurden β (95% CI) | p |
|---|---|---|---|---|---|---|---|---|---|---|---|---|---|
| PD modality | continue with CAPD | **11.54 (7.26,15.82)** | **0.00** | 1.05 (−2.96,5.06) | 0.60 | **15.95 (10.19,21.7)** | **0.00** | 4.51 (−1.47,10.49) | 0.14 | **19.23 (11.57,26.88)** | **0.00** | **22.40 (13.46,31.34)** | **0.00** |
| Age | ≤60 years | 0.09 (−4.93,5.1) | 0.97 | 2.31 (−2.39,7.01) | 0.33 | **−7.36 (−14.11,−0.61)** | **0.03** | −3.4 (−10.41,3.61) | 0.34 | −5.98 (−14.95,2.99) | 0.19 | **−12.12 (−22.59,−1.65)** | **0.03** |
| Gender | male | −1.84 (−5.89,2.21) | 0.37 | −1.79 (−5.58,2.01) | 0.35 | −2.87 (−8.32,2.58) | 0.30 | 0.77 (−4.89,6.44) | 0.78 | −4.41 (−11.66,2.84) | 0.23 | −5.96 (−14.43,2.5) | 0.17 |
| Marital status | with spouse | 1.59 (−4.44,7.62) | 0.60 | −4.71 (−10.36,0.94) | 0.10 | −1.79 (−9.9,6.33) | 0.66 | −5.49 (−13.91,2.94) | 0.20 | −2.1 (−12.89,8.69) | 0.70 | 1.6 (−10.99,14.19) | 0.80 |
| Education level | lower | −3.79 (−8.3,0.71) | 0.10 | **−5.96 (−10.18,−1.74)** | **0.00** | −5.31 (−11.37,0.75) | 0.09 | −4.53 (−10.83,1.77) | 0.16 | −0.7 (−8.76,7.35) | 0.86 | **−10.38 (−19.79,−0.98)** | **0.03** |
| Employment status | employed | −0.01 (−4.77,4.74) | 0.10 | −1.02 (−5.47,3.43) | 0.65 | 0.56 (−5.83,6.96) | 0.86 | −2.09 (−8.73,4.56) | 0.54 | −2.47 (−10.98,6.03) | 0.57 | 7.11 (−2.82,17.03) | 0.16 |
| Dependency status | self-care | −1.21 (−6.4,3.97) | 0.64 | −3.2 (−8.06,1.66) | 0.19 | 4.25 (−2.73,11.23) | 0.23 | 4.51 (−2.75,11.76) | 0.22 | 3.3 (−5.98,12.58) | 0.48 | 5.63 (−5.2,16.47) | 0.31 |
| Dialysis vintage | <4 years | 2.51 (−1.54,6.56) | 0.22 | 0.9 (−2.89,4.7) | 0.64 | 4.83 (−0.63,10.28) | 0.08 | 2.95 (−2.72,8.61) | 0.31 | 5.47 (−1.78,12.72) | 0.14 | 5.3 (−3.16,13.76) | 0.22 |
| Kt/V | ≤1.7 | −2.68 (−6.84,1.48) | 0.20 | 2.43 (−1.46,6.33) | 0.22 | 1.04 (−4.55,6.64) | 0.71 | 1.11 (−4.7,6.93) | 0.71 | 4.79 (−2.65,12.24) | 0.21 | −1.29 (−9.98,7.4) | 0.77 |
| CCI | ≤2 | 2.33 (−2.46,7.11) | 0.85 | **5.39 (1.05,9.73)** | **0.02** | 4.41 (−1.83,10.65) | 0.16 | **7.96 (1.49,14.44)** | **0.02** | 3.28 (−5.01,11.57) | 0.44 | 2.25 (−7.43,11.92) | 0.65 |
| BMI | ≤25 kg/m² | 0.85 (−3.62,5.32) | 0.34 | −1.03 (−5.51,3.46) | 0.65 | 0.38 (−6.06,6.82) | 0.90 | 2.84 (−3.85,9.53) | 0.40 | −0.25 (−8.82,8.31) | 0.95 | −0.55 (−10.55,9.44) | 0.91 |
| Albumin | <37 g/L | −1.89 (−5.93,2.15) | 0.70 | −1.08 (−5.27,3.11) | 0.61 | −4.36 (−10.38,1.66) | 0.15 | −3.46 (−9.71,2.79) | 0.28 | −7.23 (−15.23,0.77) | 0.08 | −3.04 (−12.38,6.29) | 0.52 |
| Hemoglobin | <11 g/L | 11.54 (7.26,15.82) | 0.36 | 3.65 (−0.13,7.44) | 0.06 | 2.06 (−3.38,7.5) | 0.45 | 0.63 (−5.02,6.28) | 0.83 | 1.09 (−6.13,8.32) | 0.77 | 3.28 (−5.15,11.72) | 0.44 |

**Notes.**

Values in bold indicate $P < 0.05$.

CAPD, continuous ambulatory peritoneal dialysis; APD, automated peritoneal dialysis; BMI, body mass index; PCS, physical component summary; MCS, mental component summary; KDCS, kidney disease component summary.

other reasons for higher HRQOL in APD patients in this study include remote monitoring and management of dialysis indicators conducted by specialized doctors and timely guidance from the family doctor during follow-up, which means that the technological advantages and professional supports of APD could improve the physiological and disease-specific component. This study did not find differences in MCS subscale scores, which was inconsistent with earlier cross-sectional studies that suggested that APD patients performed better in psychological states (*Mpharm et al., 2020*; *Cortés-Sanabria et al., 2013*; *Wit et al., 2001*; *Diaz-Buxo et al., 2000*). This may be due to the short observation time of this study, and the cognition and feeling of patients with chronic diseases do not change significantly within one year. Overall, it is reasonable to recommend CAPD patients convert to APD if the indications and financial circumstances permit.

We also observed the impacts of demographic, clinical, and dialysis-related characteristics on HRQOL. First, age of more than 60 years was associated with the decrease of KDCS and Burden. Age as a predictor of quality of life scores, especially in
physical aspects, was also confirmed by several previous studies (*Yang et al., 2018*; *Pan et al., 2018*; *Balasubramanian, McKitty & Fan, 2010*). Second, higher education level was associated with the decrease of MCS and Burden, which means that patients who were more educated and actively learned more about the disease might be worried about the negative outcomes. Though Yang et al. did not observe the impact of education level on the HRQOL of APD and CAPD patients (*Yang et al., 2018*), it may be caused by the well-established education system and social welfare policy in Singapore, and the fact that HRQOL is a subjective evaluation deeply influenced by social culture. Third, CCI(>2) was associated with the changes of MCS and Symptoms respectively. Previous studies explored the impact of CCI but no statistical correlation was found (*Bilgic et al., 2011*; *Yang et al., 2018*). Diseases are not mutually exclusive. Our results suggest comorbidities might have an influence on the HRQOL of ESRD patients.

There are some noteworthy implications for patients, clinical practice and health administration. For patients, this study showed that conversion to APD significantly improves HRQOL, which suggests that APD is a dialysis method worth choosing. More HRQOL evaluations should be conducted to enhance patient engagement in decision-making. For clinical practice, in addition to the objective physiological indicators, the subjective feelings of patients should also be considered during dialysis treatment. Patient-oriented medical services are worth recommending, which include involving family members and caregivers in dialysis treatment for ESRD patients, especially the older patients, and strengthening psychological support for well-educated patients. As for health administration, APD is mainly used at clinical rescue and ICU and lacks large-scale application in China. Especially in the context of COVID-19, home-based APD treatment has shown its great advantages. Priority should be given to promoting APD in the allocation of healthcare resources in China.

Some of the previously published papers found no impact of PD modality on HRQOL because the study design and the subjects' characteristics were different from our study. Former studies were mostly cross-sectional studies, and compared HRQOL between independent APD and CAPD groups. This study is a retrospective cohort study, subjects came from the same CAPD sample group at baseline, and considered the conversion of dialysis modality as an influencing factor. To the best of our knowledge, this study is the first publication reporting the impact of PD modality conversion on the HRQOL of ESRD patients. What is more, existing Asian studies mainly focus on Taiwan and Singapore patients (*Tang, Chen & Fang, 2016*; *Yang et al., 2018*). This study takes mainland China residents as the research subject. It may be that different regional, social, and cultural backgrounds will also bring differences to the quality of life. This study suggest the effectiveness of home-based APD treatment in Chinese patients using real-world data, which is an innovative approach in current Chinese literature because APD has not been widely used in mainland China.

This study also had several limitations and should be acknowledged. First, the socioeconomic status of Kunshan city is close to that of a city in a developed countries. China is a vast country with areas of different socioeconomic development, therefore the results of this study cannot be extrapolated to less developed areas in China. Second,

although we started data collection when the APD project was launched in Kunshan, the one-year follow-up might not be long enough to observe the significant changes in MCS subscale scores. Besides, even though the sample size in this study was sufficient enough to detect the effects, HRQOL is a subjective evaluation which closely related to the social and cultural background, thus, the conclusion needs further validation in other regions. Third, factors influencing the conversion of dialysis modality might also contribute to confounding bias, such as cognition and acceptance of new technology, economic level, clinical indications, etc.

## CONCLUSIONS

In summary, this study analyzed the impact of converting PD modalities on the one-year changes in HRQOL of ESRD patients in mainland China. It suggested that conversion to APD had a positive influence on the improvement of HRQOL. With the increasing use of APD in China, further studies should focus on the HRQOL of APD patients in other regions to expand the sample size and improve representativeness.

## ACKNOWLEDGEMENTS

We are grateful to all the participants who took part in this study.

### Funding

This work was supported by the National Social Science Fund of China (17ZDA078) and the Scientific Research Project of Jiangsu Provincial Health Commission (H2017073). The funders had no role in study design, data collection and analysis, decision to publish, or preparation of the manuscript.

### Grant Disclosures

The following grant information was disclosed by the authors:
The National Social Science Fund of China: 17ZDA078.
The Scientific Research Project of Jiangsu Provincial Health Commission: H2017073.

### Competing Interests

The authors declare there are no competing interests.

### Author Contributions

- Heqi Sun and Ye Zhuang conceived and designed the experiments, analyzed the data, prepared figures and/or tables, and approved the final draft.
- Lanying Gao and Yan Xiong performed the experiments, prepared figures and/or tables, and approved the final draft.
- Ningze Xu analyzed the data, authored or reviewed drafts of the paper, and approved the final draft.
- Min Yuan analyzed the data, prepared figures and/or tables, and approved the final draft.

- Jun Lu and Jianming Ye conceived and designed the experiments, authored or reviewed drafts of the paper, and approved the final draft.

## Human Ethics

The following information was supplied relating to ethical approvals (i.e., approving body and any reference numbers):

The study was approved by the Medical Research Ethics Committee of School of Public Health, Fudan University (Ethical Application Ref: IRB00002408 & FWA00002399).

## Data Availability

The raw data contains the personal information of residents. The ethical agreement permits us to use it for research, analysis and publication of the analysis and results but access to the raw data is only available upon request.

Please contact the China Research Center on Disability (cdifdu@fudan.edu.cn) to request access.

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
