# Peer review of "Impact of dialysis modality conversion on the health-related quality of life of peritoneal dialysis patients: a retrospective cohort study in China"

_PeerJ, doi:10.7717/peerj.12793_

## Round 0.1 · original submission · Major Revisions

Basic reporting:
The proposed manuscript is written in an overall good scientific style and it addresses an interesting topic in Nephrology. The topic was not extensively studied worldwide, so despite the regional character which cannot be generalized, I consider the information provided as useful and hypothesis-generating.
Experimental design:
The study methods are explained in detail and clearly. The selection of the investigated subjects seems appropriate. The results are logically presented and the Discussions and Conclusions are in line with it.
Some points need, however, further clarifications:
1. Reasons for the change in PD modality should be provided because, as authors acknowledge in the study limitations paragraph, these could have clinical meaning, so it could explain the better QoL scores. Since the study was retrospective, the change was probably dictated by medical reasons.
2. In the same line, it would be interesting to have information about the adequacy of PD before and after the change of modality. These could also account for improvements in QoL. At least daily ultrafiltration, residual diuresis, Kt/V urea should be added and included in the analysis.
3. Basic data about CKD etiology and comorbidities in the two investigated groups would be useful, as potential bias factors for QoL. Addition of Charlson Comorbidity Index is warranted.
4. Since all the investigated parameters were analyzed as categorical variables, the authors should explain why multivariate linear regression models were chosen? Logistic regression seems more appropriate for discrete parameters.
5. The authors should try to find explanations for their different results compared with previously published papers which found no impact of PD modality on QoL. Maybe something related to the subjects' characteristics?
Validity of the findings:
The findings are meaningful, but the authors should resolve the statistical methodology issues listed above.
I suggest to lessen the tone of statements in regard with the obtained results because of the study limitations (retrospective design, small sample size). Therefore, statements like "....proves the effectiveness ..." and similars ("...had significant impact...") should be avoided and replaced by "suggest", "seem", "appear" and so on.
Additional comments:
A careful revision of English language is required (for example, the phrase on lines 81-82, ending in "the most ones" is not suitable).

Reviewer 1 ·

Basic reporting

Introduction appropriately outlines the topic, provides good context for the study with adequate referencing. The results are properly reported, including tables. The discussion and conclusions are clear. I appreciate that the authors acknowledged limitations.
My only recommendation here is that the authors edit the language using professional services as English language should be improved. This is specifically the case for the Instruction section where most sentences in lines 91-96 are really difficult to follow / understand.

Experimental design

This is an original research that contributes new and interesting knowledge on the impacts of home modality conversion on the HRQOL. The research topic is within the scope of the Journal. The methodological approach is extensively described with details. The design is correct and all conducted analyses appropriate. My only two requests here are:
-the authors should provide power analyses or other justification that the sample they used was sufficient to detect the effects, I appreciate that the authors listed sample size as one of the limitations of the study, however power calculations should also be provided
-the authors conducted a lot of tests and comparisons, and I would expect some corrections to be applied to the p-values (for example Bonferroni corrections) to assure that the effects reported were not the results of so many comparisons conducted not necessary a true effect

Validity of the findings

This is a novel research with sound methodology and robust data. Conclusions are well stated, limitation are listed and the benefit to the literature as well as medical practice are stated.

Additional comments

I believe this is a very interesting study providing new and interesting knowledge on the impacts of different types of home dialysis on dialysis patients' quality of life. I recommend this study for publications, given the revisions listed above are done.

---

## Round 0.2 · Minor Revisions

I acknowledge the changes made in response to previous comments, which were adequate. I still have some minor requests:

1. On line 37, "were" is correct instead of "was" because the subjects is "...scores" which is a plural noun. Please change accordingly.

2. Please enter the statistical power analysis (its summarized result) in the Methods section of the manuscript.

3. Please enter in the Results section a phrase stating that other characteristics related to PD technique (residual diuresis and daily ultrafiltration) were not different between the two studied groups and were not correlated in univariate analysis with the HRQoL scores.

---

## Round 0.3 · accepted · Accept

The authors adequately resolved the previous suggestions. No further comments.